# Red blood cell distribution width as a risk factor for inhospital mortality in obstetric patients admitted to an intensive care unit: a single centre retrospective cohort study

Yufeng Chu,[1] Zhongshang Yuan,[2] Mei Meng,[1] Haiyan Zhou,[3] Chunting Wang,[1] Gong Yang,[4] Hongsheng Ren[1]

YC and ZY contributed equally.

[1]Department of Intensive Care Unit, Shandong Provincial Hospital Affiliated to Shandong University, Jinan, China
[2]Department of Epidemiology and Biostatistics, School of Public Health, Shandong University, Jinan, China
[3]Department of Medical Oncology, Shandong Tumour Hospital, Jinan, China
[4]Vanderbilt Epidemiology Center, Department of Medicine, Vanderbilt University Medical Center, Nashville, Tennessee, USA

**Correspondence to**
Dr Hongsheng Ren;
sdslicu@163.com, chunancy@163.com

## ABSTRACT

**Background** Red blood cell distribution width (RDW) has been shown to predict mortality in critically ill patients. To our knowledge, whether or not RDW is associated with clinical outcomes of obstetric patients requiring critical care has not been evaluated.

**Methods** This was a single centre, retrospective, observational study of obstetric patients admitted to the intensive care unit (ICU). Patients were excluded from the analysis if they had known haematological diseases or recently underwent blood transfusion. Patients who died or were discharged from the ICU within 24 hours of admission were also excluded. Patient clinical characteristics at ICU admission were retrieved from the medical charts. Multiple logistic regression was used to estimate OR and 95% CI for inhospital mortality associated with RDW. The receiver operating characteristic curve was used to examine the performance of RDW, alone or in combination with the Acute Physiology and Chronic Health Evaluation II score (APACHE II), in predicting inhospital mortality.

**Results** A total of 376 patients were included in the study. The hospital mortality rate was 5.32%. A significant association was found between baseline RDW levels and hospital mortality (OR per per cent increase in RDW, 1.31; 95% CI 1.15 to 1.49). Further adjustment for haematocrit and other potential confounders did not appreciably alter the result (p<0.001). The area under the curve (AUC) for inhospital mortality based on RDW was similar to that based on the APACHE II score (0.752 vs 0.766). A combination of these two factors resulted in substantial improvement in risk prediction, with an AUC value of 0.872 (p<0.001).

**Conclusions** The study suggests that RDW is an independent predictor for inhospital mortality among ICU admitted obstetric patients. Combining RDW and APACHE II score could significantly improve inhospital prognostic prediction among these critically ill obstetric patients.

## Strengths and limitations of this study

► To our knowledge, this is the first report of red cell distribution width (RDW) as an independent prognostic predictor of clinical outcomes in obstetric critical care patients.
► The study finding suggests that RDW, a routinely measured clinical laboratory test with high reproducibility, may have direct clinical implications and may aid the improvement of critical care for obstetric patients.
► This was a single centre study. Replication in other populations/settings is warranted.

countries.[1] [2] The use of scoring systems to assess its severity and predict mortality may help identify obstetric patients who truly require intensive care.[3] The Acute Physiology and Chronic Health Evaluation II (APACHE II) score is a predictive score for mortality that is widely used in intensive care units (ICUs). However, results from obstetric patients requiring critical care have been mixed. Some studies suggest this is a good predictor for illness severity, but other more recent studies have shown that this score has overestimated mortality risk.[4–9] Therefore, there is a real need to identify new factors in order to improve the assessment of illness severity and prediction of clinical outcomes for critically ill obstetric patients.

Red cell distribution width (RDW), a routinely measured clinical laboratory test with high reproducibility, reflects the degree of heterogeneity of erythrocyte volume.[10] RDW has been used to differentiate anaemia types over the past decades.[11] Recently, RDW has been shown to be a novel independent prognostic marker for mortality, mainly in patients with cardiovascular disease and

## INTRODUCTION

The maternal mortality ratio (remains high, despite the advances in the critical care of obstetric patients, especially in developing

strokes, as well as in critically ill patients.[12–19] However, to our knowledge, no study has directly examined the prognostic performance of RDW in obstetric patients requiring critical care.

In the study, we conducted a retrospective cohort study of ICU admitted obstetric patients to evaluate whether RDW at ICU admission was associated with inhospital mortality, and examined the performance of RDW, alone or in combination with the APACHE II score, in predicting risk for inhospital mortality.

## METHODS

The study was carried out at the Shandong Provincial Hospital Affiliated to Shandong University, Jinan, China. The hospital is a 1500 bed tertiary academic hospital with 20 ICU beds, which provides primary as well as tertiary care to an ethnically and socioeconomically diverse population within Shandong province and the surrounding region. The study was approved by the institutional review board of the hospital.

Obstetric patients consecutively admitted in the ICU for at least 24 hours from 1 January 2008 to 31 December 2013 were included in this retrospective cohort study. The requirement for patient consent was waived given the retrospective nature of this study design (ie, no direct/indirect patient care intervention, and all identifiable information was removed). Obstetric patients were defined as pregnant women or up to 6 weeks postpartum. The decision to transfer patients into the ICU was made by at least one senior critical care doctor and one senior obstetric doctor. Likewise, these doctors also made decisions to discharge patients or to transfer patients to general wards. Patients were excluded if they had known haematologic diseases (including leukaemia, thrombotic thrombocytopenic purpura and other haematological diseases) or a history of recent blood transfusion (<2 weeks). According to the APACHE II score criterion,[20] the recorded value in our study was based on the most deranged reading during each patient's initial 24 hours in the ICU, so patients who died or were discharged less than 24 hours were excluded from the study.

### Data collection

Demographic and clinical characteristics, including age, gestational age, parity, pregnancy status at admission, diagnosis at entry and during ICU stay, and length of hospital stay, were collected. Caesarean section was considered an emergency intervention. Pregnancy associated with cardiac disease included congenital and acquired heart disease during pregnancy. Diagnosis of acute kidney injury (AKI) was based on the Acute Kidney Injury Network (AKIN) criteria.[21] AKI was defined as an absolute increase in serum creatinine of ≥0.3 mg/dL, a percentage increase in serum creatinine of ≥50% (1.5-fold from baseline) within 48 hours or a reduction in urine output (documented oliguria of <0.5 mL/kg per hour for more than 6 hours).[21] APACHE II scores were

calculated using the worst value of 12 acute physiological variables within 24 hours of presentation. These variables included temperature, blood pressure, heart rate, respiratory rate, arterial oxygenation, arterial pH, serum sodium, serum potassium, serum creatinine, haematocrit, white blood cell count and Glasgow Coma score. If the patient was sedated at the time of ICU admission, the last Glasgow Coma Scale obtained prior to sedation was collected. RDW, haemoglobin level, haematocrit and mean corpuscular volume of all patients included in this study were determined at ICU admission using a Beckman Coulter LH-750 Haematology Analyser (Beckman Coulter Inc, Fullerton, California, USA) as part of the complete blood cell count. Normal reference range for RDW in the hospital laboratory is between 10.9% and 15.4%. In addition, we obtained the number of birth (from 1 January 2008 to 31 December 2013) from the hospital database to calculate ICU admission rate in the hospital.

### Study outcomes

All patients were followed-up during hospitalisation. The primary endpoint of the study was inhospital mortality. The primary predictor of interest was RDW measured at ICU admission.

### Statistical analysis

All continuous variables are presented as mean±SD or medians (interquartile range), as appropriate. Categorical data are summarised as number or percentage. Patient characteristics across tertiles of RDW were compared using analysis of variance or the Kruskal–Wallis test for continuous variables and $\chi^2$ or Fisher's exact test for categorical variables. Logistic regression analysis was conducted to estimate OR and 95% CI for inhospital mortality associated with RDW and other clinical parameters after adjustment for potential confounding factors. The receiver operating characteristic (ROC) curve was used to examine the performance of RDW, alone or in combination with other clinical parameters such as APACHE II score, in predicting inhospital mortality. The curve represents a sensitivity plot versus 1–specificity. The area under the curve (AUC) was derived from the ROC curve, and the Youden Index was adopted to define the optimal cut-off value.[22] We also constructed an ROC curve for the combined APACHE II score and RDW results for predicting inhospital mortality according to the weighted sum formula derived from multivariate logistic regression: logit (mortality)=0.162 × APACHE II score +0.203 × RDW − 7.503; where logit (mortality) is the logarithm of the odds of a critically ill obstetric patient dying in the ICU. Differences between the AUC were detected by Delong's test, which was a non-parametric approach and could generate an estimated covariance matrix by using the theory on generalised U-statistics.[23] Two sided p values <0.05 were considered statistically significant. All analyses were performed with R software (http://www.cran.r-project.org/).

## RESULTS
### Population
A total o f 20 570 births were reported from 1 January 2008 to 31 December 2013 in Shandong Provincial Hospital, among which 447 obstetric patients were admitted to the ICU. The ICU admission rate was 21.73 per 1000 births in the hospital; Eight patients who were diagnosed with haematological disease and 59 patients who received red blood cell transfusion within 2 weeks were excluded. Two patients who died within 24 hours after ICU admission were also excluded. Also, two patients were excluded for missing data. Thus, in the final analysis, 376 patients were included in the study.

### Association between RDW and hospital mortality
A total of 20 deaths occurred in this cohort during the study period. Heart failure was the major cause of death in the cohort (n=8; 40.0%), followed by acute fatty liver of pregnancy (n=5; 25%), postpartum haemorrhage (n=2; 10%), haemorrhagic shock caused by liver tumour rupture (n=1; 5%), HELLP syndrome (n=1; 5%), acute pulmonary embolism (n=1; 5%), stroke (n=1; 5%) and liver failure caused by severe hepatitis B (n=1; 5%). The inhospital mortality was 5.32%.

RDW ranged from 10.7% to 24.0% (median 14.8%; IQR 13.4% to 16.0%). Table 1 shows a comparison of patient clinical characteristics across tertiles of RDW. RDW levels were inversely associated with levels of haemoglobin, mean corpuscular volume and haematocrit but positively associated with APACHE II score and inhospital mortality. No difference was found in distributions of age, gestational weeks, primary reasons for ICU admission, AKI morbidity or the total length of stay in hospital across RDW tertiles.

### Regression analysis for inhospital mortality
Univariate logistic regression analysis demonstrated that patients with a higher RDW, higher APACHE II scores,

**Table 1** Baseline clinical and laboratory characteristics by tertile of red cell distribution width at critical care initiation

| | Tertile I | Tertile II | Tertile III | p Value |
|---|---|---|---|---|
| **RDW (%)** | 10.7–13.9 | 13.9–15.6 | 15.6–20.4 | <0.001 |
| **N** | 131 | 126 | 119 | - |
| **Age (years)** | 29.6±4.3 | 29.3±5.6 | 29.6±5.9 | 0.84 |
| **Gestational age (weeks)** | 35 (32–37) | 36 (33–39) | 36 (32–38) | 0.327 |
| **Primary diagnosis to ICU admission (n (%))** | | | | |
| Hypertensive disorder of pregnancy | 50 (38.17) | 55 (43.65) | 52 (43.70) | 0.417 |
| HELLP syndrome | 12 (9.16) | 10 (7.94) | 8 (6.72) | 0.322 |
| Acute fatty liver of pregnancy | 10 (7.63) | 8 (6.35) | 7 (5.88) | 0.485 |
| Obstetric sepsis | 5 (3.82) | 4 (3.17) | 3 (2.52) | 0.657 |
| Cardiovascular disease | 42 (32.06) | 31 (24.60) | 26 (21.85) | 0.162 |
| Gastrointestinal disease | 2 (1.53) | 5 (3.97) | 4 (3.36) | 0.476 |
| Stroke | 2 (1.53) | 3 (2.38) | 0 (0) | 0.332 |
| Pulmonary embolism | 1 (0.76) | 0 (0) | 1 (0.84) | 0.766 |
| Others | 7 (5.34) | 10 (7.94%) | 18 (15.13) | 0.142 |
| **Haemoglobin (g/L)** | 109.5±19.7 | 100.1±20.7 | 92.5±24.1 | <0.001 |
| **MCV (fL)** | 89.8±5.1 | 86.7±6.0 | 83.6±11.4 | <0.001 |
| **HCT (%)** | 32.6±5.8 | 30.5±6.3 | 28.8±6.9 | <0.001 |
| **APACHE II score (points)** | 2 (2–6) | 5 (3–6) | 10 (6–22) | 0.017 |
| **AKI (n %)** | 14 (10.69) | 17 (13.49) | 14 (11.76) | 0.207 |
| **TLSH (days)** | 8 (7–12) | 8 (6–13) | 8 (7–12) | 0.203 |
| **Hospital mortality (n (%))** | 0 (0) | 5 (3.97) | 15 (12.61) | <0.001 |

Values are presented as mean±SD or number (%).
AKI, acute kidney injury; APACHE II score, Acute Physiology and Chronic Health Evaluation II score; HCT, haematocrit; MCV, mean corpuscular volume; RDW, red cell distribution width; TLSH, total length of stay in hospital.

**Table 2** Univariate ORs of variables for predicting inhospital mortality

| Variable | OR | 95% CI | p Value |
|---|---|---|---|
| Age (years) | 1.048 | 0.969 to 1.133 | 0.244 |
| Haemoglobin (g/L) | 0.997 | 0.977 to 1.017 | 0.763 |
| MCV (fL) | 0.962 | 0.919 to 1.006 | 0.098 |
| HCT (%) | 0.997 | 0.934 to 1.064 | 0.929 |
| APACHE II score (points) | 1.192 | 1.124 to 1.265 | <0.001 |
| AKI (%) | 16.61 | 6.580 to 42.014 | <0.001 |
| TLSH (days) | 0.803 | 0.691 to 0.933 | 0.004 |
| Gestational age (weeks) | 1.023 | 0.920 to 1.138 | 0.677 |
| RDW (%) | 1.309 | 1.150 to 1.489 | <0.001 |

AKI, acute kidney injury; APACHE II score, Acute Physiology and Chronic Health Evaluation II score; HCT, haematocrit; MCV, mean corpuscular volume; RDW, red cell distribution width; TLSH, total length of stay in hospital.

higher AKI morbidity and longer TLSH days had significantly greater death hazards (table 2). Every 1% increase in RDW was associated with a 31% increase in the risk, with an OR of 1.31 (95% CI 1.15 to 1.49).

Multivariate logistic regression analysis revealed that RDW, AKI and APACHE II scores were independent predictors of inhospital mortality (table 3). The association of RDW and inhospital mortality remained significant after adjusting for age, haemoglobin, MCV, haematocrit, APACHE II score and AKI. RDW was a significant outcome predictor, which was independent of APACHE II score.

An ROC curve was drawn to evaluate the value for RDW and APACHE II scores in predicting mortality (figure 1). RDW and APACHE II score were equally sensitive in prognostic prediction, with an AUC of 0.766 (95% CI 0.705 to 0.826) for the APACHE II score and 0.752 (95% CI 0.684 to 0.862) for RDW. The optimal cut-off value of the APACHE II score for predicting mortality was 5 points, which yielded sensitivity and specificity of 90.9% and 60.7%, respectively. The optimal cut-off value of RDW was 16.1%, which resulted in sensitivity and specificity values of 68.2% and 77.9%, respectively. We further combined RDW and the APACHE II score to draw a third ROC curve, as shown in figure 1, yielding much greater discriminatory capacity for inhospital mortality, with an AUC of 0.872. As shown in the multiple logistic regression analysis (table 3), AKI was also significantly associated with inhospital mortality. However, no appreciable improvement in prognostic performance was observed when further incorporating AKI into the prediction model. AUC derived from all three variables was 0.884. The Delong's z statistic was −0.668 and the p value was 0.504.

## DISCUSSION

The main finding of our study was that RDW was independently associated with inhospital mortality in obstetric critical care patients. The association remained significant after adjusting for APACHE II score, haemoglobin levels, haematocrit and mean corpuscular volume. The study revealed the first evidence that the prognostic performance of RDW for inhospital mortality among ICU admitted obstetric patients was similar to APACHE II score, a widely accepted predictor of clinical outcomes in critically ill patients. A combination of RDW and APACHE II score provided even greater predictive power than either alone.

RDW has been associated with all-cause mortality in critically ill patients.[16 24 25] For the first time, we specifically evaluated the association among obstetric patients requiring critical care. We found that RDW was independently associated with inhospital mortality in these patients. Patients were excluded if they had a history of recent blood transfusions because RDW could be increased in anaemia or after blood transfusions.[11 26] In the present study, a higher level of RDW was associated with inhospital mortality even after adjusting for

**Table 3** Independent predictors of inhospital mortality by multivariate logistic regression analysis

| Variable | OR | 95% CI | p value |
|---|---|---|---|
| APACHE II (points) | 1.189 | 1.071 to 1.319 | 0.001 |
| AKI (%) | 23.784 | 6.129 to 92.296 | <0.001 |
| RDW (%) | 1.401 | 1.156 to 1.697 | 0.001 |

Note, variables in the model included age, haemoglobin, mean corpuscular volume, haematocrit, APACHE II score, AKI, TLSH, gestational age and RDW.
Due to the high correlation between haemoglobin and haematocrit, haemoglobin was first regressed on haematocrit.
AKI, acute kidney injury; then placed the residual and haematocrit in the multivariate regression. APACHE II score, Acute Physiology and Chronic Health Evaluation II score; RDW, red cell distribution width.

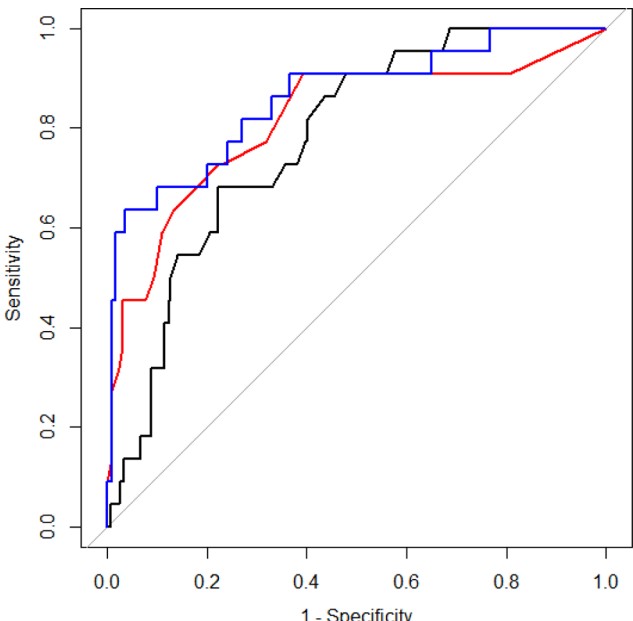

**Figure 1** Receiver operating characteristic curve for Acute Physiology and Chronic Health Evaluation II score, red cell distribution width and the combination of both in predicting hospital mortality.

haemoglobin and haematocrit. In addition, the independent association between RDW and hospital mortality was not eliminated after further adjusting for other potential confounders, such as MCV, haemoglobin, haematocrit, APACHE II score, gestational age and AKI. Our findings indicated that combining RDW and APACHE II score performed better than APACHE II score alone in predicting inhospital mortality in critically ill obstetric patients.

The APACHE II scoring system developed in 1985 has shown a positive correlation with hospital mortality, and was one of the most common models used for evaluating the severity of a disease in critically ill patients.[20] The sensitivity and specificity of the APACHE II score was evaluated with the use of ROC curve analysis in the present study. In accordance with previous studies,[27–29] the APACHE II score was demonstrated to have a moderate discriminative ability to predict inhospital mortality (AUC=0.766), similar to that for RDW (AUC=0.752). Combining RDW with APACHE II score significantly improved prognostic performance (AUC=0.872). Similarly, another study by Wang et al[16] also revealed that adding RDW to APACHE II score significantly improved the prognostic reliability of APACHE II score in identifying critically ill patients. Therefore, RDW, a simple, inexpensive and widely available clinical test as part of the complete blood count may have significant clinical implications for determining prognosis in critically ill obstetric patients. AKI was also significantly associated with inhospital mortality. However, no significant improvement in prognostic performance was observed when further incorporating AKI into the prediction model.

The pathophysiological mechanism underlying the association of higher RDW with worse outcomes in critically ill

obstetric patients remains unclear. Generally, the increase in RDW reflects either impaired erythropoiesis, abnormal red blood cell survival or both. Metabolic abnormalities such as shortening of telomere length, poor nutritional status,[30 31] inflammation,[32–34] oxidative stress,[35 36] dyslipidaemia, hypertension, erythrocyte fragmentation or alteration of erythropoietin function might contribute to RDW increase.[37] In normal parturition, the increase in RDW might be related to stimulus induced reticulocytosis in the last few weeks.[30] This might not be the case in our study where, even if five patients with spontaneous onset of labour were excluded, RDW remained a significant predictor of mortality.con

In this present study, the admission rate of 21.73 per 1000 births was relatively high,[27 38] and the mean APACHE II score was relatively low compared with other studies.[7 16 39] The lack of a high dependency unit as a referral centre in local areas for complicated pregnancies could partially explain these differences.

This study had several limitations. First, this work did not evaluate potential changes in RDW over time which may provide additional prognostic information. Second, only the APACHE II score was used to predict mortality in obstetric patients, which was limited by spontaneous improvement after delivery in peripartum women[40–46] and inconsistent results.[47 48] Finally, the study was conducted in a single centre. It is nto examine the association in different study settings/populations with a larger sample size.

In summary, the study suggested that RDW may be an independent predictor for in-hospital mortality in obstetric patients requiring clinical care. Combining RDW and APACHE II score significantly improved prognostic assessment among critically ill obstetric patients, which may have direct clinical implications and may aid the improvement of critical care for obstetric patients.

**Contributors** YC and HR conceived of the study, and participated in its design and coordination. MM, HZ and CW extracted data and participated in the study design. YC, ZY performed the statistical analysis and drafted the manuscript. GY interpreted the data and critically revised the manuscript. All authors read and approved the final manuscript.

**Funding** `The study was funded by the National Natural Science Foundation of China (NSFC 81200238), the Research Foundation for Excellent Young and Middle-aged Scientists of Shandong Province (BS2011YY043, BS2011YY052),the Young Scholars Program of Shandong University (2016WLJH23), the clinical medicine science and technology innovation plan of Jinan Bureau of Science and Technology(201602164), the Shandong Provincial Department of Science and Technology research and development plan (2016GSF201052) and the Natural Science Foundation of Shandong Province of China (ZR2013HM062).

**Competing interests** None declared.

**Patient consent** The requirement for patient consent was waived because this study did not affect the patient's clinical care, and all protected health information was deleted.

**Ethics approval** The study was approved by the institutional review board of the Provincial Hospital Affiliated to Shandong University, China.

**Provenance and peer review** Not commissioned; externally peer reviewed.

**Data sharing statement** No additional unpublished data are available.

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
