## [Reviewer comments · BMJ Open]

ARTICLE DETAILS

TITLE (PROVISIONAL)	Red blood cell distribution width as a risk factor for in-hospital mortality in obstetric patients admitted to Intensive care unit – a single center, retrospective cohort study.
AUTHORS	Chu, Yufeng; Yuan, Zhongshang; Meng, Mei; Zhou, Haiyan; Wang, Chunting; Yang, Gong; Ren, Hongsheng

VERSION 1 - REVIEW

REVIEWER	Yusuke Uemura Cardiovascular Center, Anjo Kosei Hospital, Japan
REVIEW RETURNED	15-Jun-2016

GENERAL COMMENTS	The authors demonstrated in the manuscript; “Red cell distribution width improves prognostic performance of the acute physiology and chronic health evaluation II score in obstetric patients admitted to intensive care unit”, that RDW is an independent predictor for ICU mortality and that adding RDW to APACHE-II scores improved the prognostic performance. The findings would be interesting, and have several concerns and questions as follows to be addressed. 1. As you mentioned in discussion part, ICU admission rate was relatively high and APACHE-II score was relatively low in this study. These might affect the maternal mortality ratio. However, there are large disparities between countries, but also within countries in maternal mortality ratio. You should clarify maternal mortality ratio in your institution.2. The primary endpoint of this study was ICU mortality. However, admission or discharge criteria vary among institutions, regions, and countries, introducing heterogeneity, especially in obstetric patients. Furthermore, in perinatal care, maternal mortality is as important as ICU mortality. Are both APACHE-II scores and RDW levels also associated with maternal mortality in this population?3. In primary diagnosis to ICU admission of this study, the rate of cardiovascular disease seems to be high. What does “cardiovascular disease” mean? The definition of “cardiovascular disease” should be included in method part.4. ICU mortality was 5.32%. What were the causes of death in these patients?5. In multivariate logistic regression analysis, AKI is a strong predictor for ICU mortality. Is prognostic accuracy of ICU mortality improved by the combination of RDW levels with not only APACHE-II but also the incidence of AKI?
---

REVIEWER	I.A.Meynaar Intensive Care Unit HagaHospital The Netherlands
REVIEW RETURNED	15-Oct-2016

GENERAL COMMENTS	In this manuscript the authors report that in critically ill obstetric patients, the RDW is independently associated with increased ICU mortality. The manuscript is written clearly and concisely. I have only a few minor concerns. Pg 4, L 17: APACHE II is widely used in ICUs but not in the general population. Pg 5, L 49: Why were patients who died or were discharged within 24 hours excluded? The explanation given by the authors, that these data were difficult to obtain, is insufficient. Why are these data difficult to obtain? It would be more logical if patients who were in the ICU for less than 8 hours were excluded, because APACHE II requires at least 8 hours of observation. Pg 6, L54: What is the use of following up on patients during their entire hospitalization of the study endpoint is ICC mortality. Also, please note that APACHE II predicts hospital mortality, not ICU mortality. Pg 9, L45: How do you explain that confidence intervals for ROC AUC for APACHE II and APACHE II + RDW overlap but you state that there is a significant difference between the two? Pg 10, L19: Why do you conclude that the results are true for the Chinese population only? What has race or nationality got to do with it? I am sure that a similar study in Europe would not report that the results hold true for a European, English or Caucasian population. Unless you have a clue that race or nationality are involved of course.
---

VERSION 1 – AUTHOR RESPONSE

Reviewer: 1

Reviewer Name: Yusuke Uemura

Institution and Country: Cardiovascular Center, Anjo Kosei Hospital, Japan

Competing Interests: None declared

The authors demonstrated in the manuscript; “Red cell distribution width improves prognostic performance of the acute physiology and chronic health evaluation II score in obstetric patients admitted to intensive care unit”, that RDW is an independent predictor for ICU mortality and that adding RDW to APACHE-II scores improved the prognostic performance. The findings would be interesting, and have several concerns and questions as follows to be addressed.

1. As you mentioned in discussion part, ICU admission rate was relatively high and APACHE-II score was relatively low in this study. These might affect the maternal mortality ratio. However, there are large disparities between countries, but also within countries in maternal mortality ratio. You should clarify maternal mortality ratio in your institution.

Response: Thanks for your comments. As you mentioned, there are large disparities of maternal mortality ratio (MMR) between countries. China, as a developing country, has a higher MMR than developed countries. Based on your suggestion, I have added some contents on MMR in our

institution in discussion (P13, L282-286) just as follows: There were 20570 births from 2008 to 2013 in Shandong Provincial Hospital and 447 ICU admissions. A total of 28 deaths occurred in the hospital during the 6-year study period resulting in a maternal mortality rate of 136 per 100,000 births. There were 22 deaths occurred in the ICU, and the others occurred in emergency room (n=2) and maternity ward (n=4).

2. The primary endpoint of this study was ICU mortality. However, admission or discharge criteria vary among institutions, regions, and countries, introducing heterogeneity, especially in obstetric patients. Furthermore, in perinatal care, maternal mortality is as important as ICU mortality. Are both APACHE-II scores and RDW levels also associated with maternal mortality in this population?

Response: Thanks for your valuable suggestions. It is indeed to clarify the maternal mortality in the manuscript. According to the rule of our department, all ICU patients will be followed up at 1, 6, and 12 months after discharge. As a result, of the total of 376 obstetric patients who were admitted to ICU, all the death occurred in the ICU. That is to say, there was no death outside of hospital and ICU. Thus, the maternal mortality is equal to ICU mortality in this population. I have modified it and added the corresponding comments in the revision (P8, L174-177).

3. In primary diagnosis to ICU admission of this study, the rate of cardiovascular disease seems to be high. What does "cardiovascular disease" mean? The definition of "cardiovascular disease" should be included in method part.

Response: Thanks for your guidance. The "cardiovascular disease" should be "pregnancy associated with cardiac disease", and the cardiac disease included congenital and acquired heart disease. To make it more clear, we have added these explanations in the revised manuscript (P6, L117-119)

4. ICU mortality was 5.32%. What were the causes of death in these patients?

Response: Thanks. It has been added in results as follows: Heart failure was the major cause of death in the cohort (n=8; 40.0%), followed by acute fatty liver of pregnancy (n=5; 25%), postpartum hemorrhage (n=2), hemorrhagic shock caused by liver tumor rupture (n=1), HELLP syndrome (n=1), acute pulmonary embolism (n=1), stroke (n=1), and liver failure caused by severe hepatitis B (n=1) (P9, L179-184).

5. In multivariate logistic regression analysis, AKI is a strong predictor for ICU mortality. Is prognostic accuracy of ICU mortality improved by the combination of RDW levels with not only APACHE-II but also the incidence of AKI?

Response: Thanks for your kind reminder. Following your instructions, we have re-conducted the ROC curve analysis using all three variables including RDW, APACHE-II and AKI. Adding AKI improved the AUC from 0.872 to 0.884, however, when we compare them using Delong's test, no significance can be found. It was added in results as below: "As in multivariate logistic regression analysis, AKI was another strong predictor for hospital mortality. We conducted the ROC curve analysis using all three variables including RDW, APACHE-II and AKI. Adding AKI improved the AUC from 0.872 to 0.884 ($P>0.05$), however, when we compared them by Delong's test, no significance can be found." (P10, L218-222).

In discussion, revised as follows: "However, no significant improvement on prognostic performance was observed when AKI was associated with RDW and APACHE-II score, even though AKI was another strong predictor for hospital mortality in multivariate logistic regression analysis. The reason was not clear, and further study would be needed." (P12, L266-270)

Reviewer: 2

Reviewer Name: I.A.Meynaar

Institution and Country: Intensive Care Unit, HagaHospital, The Netherlands

Competing Interests: None declared

In this manuscript the authors report that in critically ill obstetric patients, the RDW is independently associated with increased ICU mortality. The manuscript is written clearly and concisely. I have only a few minor concerns.

Pg 4, L 17: APACHE II is widely used in ICUs but not in the general population.

Response: Thanks for your attention. We have revised in manuscript as you mentioned “APACHE II is widely used in ICUs” (P4, L67).

Pg 5, L 49: Why were patients who died or were discharged within 24 hours excluded? The explanation given by the authors, that these data were difficult to obtain, is insufficient. Why are these data difficult to obtain? It would be more logical if patients who were in the ICU for less than 8 hours were excluded, because APACHE II requires at least 8 hours of observation.

Response: Thanks for your guidance. It is indeed necessary to added some explanations. According to the APACHE II score criterion of Knaus WA, 1985(CCM.1985 Oct;13(10):818-29), the recorded value in our study is still based on the most deranged reading during each patient’s initial 24h in an ICU, so we excluded the patients who died or were discharged less than 24 hours. We have added these explanations in the revision (P5, L104-107).

Pg 6, L54: What is the use of following up on patients during their entire hospitalization of the study endpoint is ICC mortality. Also, please note that APACHE II predicts hospital mortality, not ICU mortality.

Response: Thanks for your correction. I have modified it in revised manuscript. It is indeed to clarify the hospital mortality in the manuscript. All patients were followed up during hospitalization. As a result, of the total of 376 obstetric patients who were admitted to ICU, all the death occurred in the ICU. That is to say, there was no death outside of hospital and ICU. Thus, the hospital mortality is equal to ICU mortality in this population.

Pg 9, L45: How do you explain that confidence intervals for ROC AUC for APACHE II and APACHE II + RDW overlap but you state that there is a significant difference between the two?

Response: Thanks for your insightful comments. If two or more empirical curves are constructed based on tests performed on the different individuals, then as you mentioned, there should be no significant difference between two ROCs AUC if the two confidence intervals overlap. In our manuscript, the two ROC curves are constructed based on tests performed on the same individuals, and the correlation between the curves must be taken into account. We adopted Delong’s test to detect their difference, which is a nonparametric approach and can generate an estimated covariance matrix by using the theory on generalized U-statistics. That is, when the correlation had been considered, our results show the significant difference. Based on your valuable suggestions, we have added detailed explanation of Delong’s test in the revision (P8, L158-160).

Pg 10, L19: Why do you conclude that the results are true for the Chinese population only? What has race or nationality got to do with it? I am sure that a similar study in Europe would not report that the results hold true for a European, English or Caucasian population. Unless you have a clue that race or nationality are involved of course.

Response: Thanks for your valuable comments. As you said, the result may be not only limited in the Chinese people. I have modified it in revised manuscript as “RDW is an independent predictor for ICU mortality in obstetric critical care patients”. (P3, L46-47).

VERSION 2 – REVIEW

REVIEWER	I.A.Meynaar ICU HagaZiekenhuis The Hague Netherlands
REVIEW RETURNED	27-Dec-2016

GENERAL COMMENTS	To my disappointment the authors have not answered my previous remarks in a point by point manner, neither did the authors provide a
--

	version of the manuscript that clearly tracks changes and since I did not save my remarks, I am obliged to judge the whole manuscript as if it is a new manuscript. The message is clear though; RDW is an independent predictor of mortality in obstetric critically ill patients in this retrospective cohort study. I have some minor remarks that I am sure can easily be answered. Title Pg 1, ln 2: The title does not give a clue on the kind of study this is (a retrospective cohort study). Instead the study is called a study on the clinical usefulness of RDW, but the manuscript does not study clinical usefulness but statistical significance. If the authors want to give the manuscript a meaningful name I would suggest: RDW as a risk factor in obstetric patients admitted to the ICU – a retrospective cohort study. Abstract Pg 2, ln 29 and 37. Did you include 376 or 447 patients? Pge 2 line 33 and 40. You performed logistic regression, but you report hazard ratio's not odds ratios. Please explain or correct. Methods The term maternal mortality is introduced in this revision. Please define. What is the difference between maternal mortality and hospital mortality? Pg 7, ln 134. What is the use of 1, 6 and 12 month follow-up for this study? Results Pg 8, ln 167-173. It is confusing to state that 447 patient were initially enrolled. Better to state that 447 were considered or eligible, counting the excluded patients to end up with 376 included patients. Pg 8, ln 168. The number of 21.73 per 1000 births is not a result of this study (you did not count the number of births in this study. It is worth mentioning though, but I would suggest putting it in the methods were you describe the setting. And: do you mean per 1000 births in the hospital? In the region? Discussion Pg 12 ln 257-258. AUC for APACHE II of 0.766 is called strong, while AUC of RDW of 0.752, which is only marginally less, is called moderate. This is contradictory. Pg 13, ln 283-292. These are new results, which appear freshly in the discussion. They belong in the introduction, the methods or the result section. The conclusion on the lack of a high dependency unit is not supported by the results. Referring to the proposed title: the clinical usefulness of RDW is not discussed in the discussion.
--	---

VERSION 2 – AUTHOR RESPONSE

We sincerely appreciate your effort in reviewing our manuscript. Your comments are very constructive and we have revised it accordingly. Responses to your specific comments are given below.

To my disappointment the authors have not answered my previous remarks in a point by point manner, neither did the authors provide a version of the manuscript that clearly tracks changes and

since I did not save my remarks, I am obliged to judge the whole manuscript as if it is a new manuscript.

Response: We apologize very much for this and thanks for your patience. The reason is that we have misunderstood the authors' instructions for revision submission and failed to upload the point by point response to your comments. Below are the new responses to your new comments, followed by the previous responses to your previous remarks.

Title

Pg 1, Ln 2: The title does not give a clue on the kind of study this is (a retrospective cohort study). Instead the study is called a study on the clinical usefulness of RDW, but the manuscript does not study clinical usefulness but statistical significance. If the authors want to give the manuscript a meaningful name. I would suggest: RDW as a risk factor in obstetric patients admitted to the ICU – a retrospective cohort study.

Response: Thanks for your valuable suggestions. Following your comments, the title has been revised as "RDW as a risk factor in obstetric patients admitted to the ICU- a single center, retrospective cohort study".

Abstract

1. Pg 2, Ln 29 and 37. Did you include 376 or 447 patients?

Response: Thanks for your comments. There were 447 ICU admissions during study and some patients were excluded for some reasons, so actually, a total of 376 patients were included in this present study. To be more clear, I deleted this sentence in methods "A total of 447 consecutive obstetric patients were included"

and keep the sentence "A total of 376 patients were included in this present study" only in the result section.

2. Pge 2 line 33 and 40. You performed logistic regression, but you report hazard ratio's not odds ratios. Please explain or correct.

Response: Thanks for your attention. It should be odds ratio, we have checked the whole manuscript carefully and changed all hazard ratio to be odds ratio.

Methods:

1. The term maternal mortality is introduced in this revision. Please define. What is the difference between maternal mortality and hospital mortality?

Pg 7, Ln 134. What is the use of 1, 6 and 12 month follow-up for this study?

Response: Thanks for pointing this out. Actually, we have misunderstood the meaning of the term "maternal mortality", it should be the number of maternal deaths per 100 000 live births. It is indeed a big mistake that I thought maternal mortality means the total mortality during pregnancy and perinatal period before. We really appreciate your comments and help us to improve the manuscript. We have revised the mortality as hospital mortality, and as you mentioned, the follow-up information seems a little redundant, and we have deleted them in the revision.

Results 1. Pg 8, Ln 167-173. It is confusing to state that 447 patient were initially enrolled. Better to state that 447 were considered or eligible, counting the excluded patients to end up with 376 included patients. Response: Thanks for your comments. It is really confused about 447 and 376. According to your suggestion, it was revised as "There were 447 eligible obstetric patients who were admitted to

ICU during the study period. ICU admission rate was 21.73 per 1,000 births in the hospital. Eight patients who were diagnosed with hematologic disease and 59 patients who received red blood cell transfusion within two weeks were excluded. Two patients who died within 24 hours after ICU admission were also excluded. Two patients were excluded for missing data. Thus, in the final analysis, 376 patients were included in this study.”

2. Pg 8, In 168. The number of 21.73 per 1000 births is not a result of this study (you did not count the number of births in this study. It is worth mentioning though, but I would suggest putting it in the methods where you describe the setting. And: do you mean per 1000 births in the hospital? In the region?

Response: Thanks for your kind reminder. We have added “In addition, we got the number of birth (from January 1, 2008 to December 31, 2013) from hospital database in order to count ICU admission rate in the hospital” in the method section. And here we mean that ICU admission rate was 21.73 per 1,000 births in the hospital.

Discussion

1. Pg 12 In 257-258. AUC for APACHE II of 0.766 is called strong, while AUC of RDW of 0.752, which is only marginally less, is called moderate. This is contradictory.

Response: Thanks for your correction. Just as you mentioned, AUC of APACHE II and RDW is very closed, but “strong and moderate” makes a great difference. It was revised as “ In accordance with previous studies²⁷⁻²⁹, the APACHE-II score was also demonstrated to have a good ability to predict hospital mortality (AUC=0.766) in the present study”.

2. Pg 13, In 283-292. These are new results, which appear freshly in the discussion. They belong in the introduction, the methods or the result section. The conclusion on the lack of a high dependency unit is not supported by the results.

Response: Thanks. After carefully checking, we have put them into the result section.

3. Referring to the proposed title: the clinical usefulness of RDW is not discussed in the discussion.

Response: Please see our response to your Title question above.

Responses to the reviewer's previous remarks

In this manuscript the authors report that in critically ill obstetric patients, the RDW is independently associated with increased ICU mortality. The manuscript is written clearly and concisely. I have only a few minor concerns.

Pg 4, L 17: APACHE II is widely used in ICUs but not in the general population.

Response: Thanks for your attention. We have revised in manuscript as you mentioned “APACHE II is widely used in ICUs” (P4,L67).

Pg 5, L 49: Why were patients who died or were discharged within 24 hours excluded? The explanation given by the authors, that these data were difficult to obtain, is insufficient. Why are these data difficult to obtain? It would be more logical if patients who were in the ICU for less than 8 hours were excluded, because APACHE II requires at least 8 hours of observation.

Response: Thanks for your guidance. It is indeed necessary to add some explanations. According to the APACHE II score criterion of Knaus WA, 1985(CCM.1985 Oct;13(10):818-29), the recorded value in our study is still based on the most deranged reading during each patient's initial 24h in an ICU, so

we excluded the patients who died or were discharged less than 24 hours. We have added these explanations in the revision (P5, L104-107).

Pg 6, L54: What is the use of following up on patients during their entire hospitalization of the study endpoint is ICC mortality. Also, please note that APACHE II predicts hospital mortality, not ICU mortality.

Response: Thanks for your correction. I have modified it in revised manuscript. It is indeed to clarify the hospital mortality in the manuscript. All patients were followed up during hospitalization. As a result, of the total of 376 obstetric patients who were admitted to ICU, all the death occurred in the ICU. That is to say, there was no death outside of hospital and ICU. Thus, the hospital mortality is equal to ICU mortality in this population.

Pg 9, L45: How do you explain that confidence intervals for ROC AUC for APACHE II and APACHE II + RDW overlap but you state that there is a significant difference between the two?

Response: Thanks for your insightful comments. If two or more empirical curves are constructed based on tests performed on the different individuals, then as you mentioned, there should be no significant difference between two ROCs AUC if the two confidence intervals overlap. In our manuscript, the two ROC curves are constructed based on tests performed on the same individuals, and the correlation between the curves must be taken into account. We adopted Delong's test to detect their difference, which is a nonparametric approach and can generate an estimated covariance matrix by using the theory on generalized U-statistics. That is, when the correlation had been considered, our results show the significant difference. Based on your valuable suggestions, we have added detailed explanation of Delong's test in the revision (P8, L158-160).

Pg 10, L19: Why do you conclude that the results are true for the Chinese population only? What has race or nationality got to do with it? I am sure that a similar study in Europe would not report that the results hold true for a European, English or Caucasian population. Unless you have a clue that race or nationality are involved of course.

Response: Thanks for your valuable comments. As you said, the result may be not only limited in the Chinese people. I have modified it in revised manuscript as "RDW is an independent predictor for ICU mortality in obstetric critical care patients". (P3, L46-47).

VERSION 3 – REVIEW

REVIEWER	I.A.Meynaar ICU HagaZiekenhuis The Hague Netherlands
REVIEW RETURNED	23-Feb-2017

GENERAL COMMENTS	I would suggest another check by a native speaker but have no further questions or remarks.
---